# Integrating Sustainable Stormwater Management in Urban Planning: Ways Forward towards Institutional Change and Collaborative Action

**Anna Bohman \*, Erik Glaas and Martin Karlson**

Department of Thematic Studies—Environmental Change, Centre for Climate Science and Policy Research, Linköping University, SE-581 83 Linköping, Sweden; erik.glaas@liu.se (E.G.); martin.karlson@liu.se (M.K.)
\* Correspondence: anna.bohman@liu.se; Tel.: +46-13-285879

**Abstract:** Climate change impacts, ageing infrastructure and the increasing imperviousness of cities all raise enormous challenges to and call for new ways of planning for sustainable urban stormwater management. Especially, closer collaboration among a diverse set of actors involved has been pointed to as critical to enable the development of holistic and flexible approaches. However, the shift towards inclusive forms of planning has been slow, and characterized by technical and institutional lock-ins. Against this background, this study scrutinizes the challenges and developments perceived as central for improving stormwater planning, and analyzes how formal and informal institutional change could contribute to enhancing sustainability in this sector. Building on an analysis of data from workshops, interviews and a survey with Swedish planners and water managers, we suggest new strategies for integrating stormwater concerns into planning processes, overcoming silo structures, fostering cocreation cultures, and securing the continuation and implementation of stormwater management through various planning stages.

**Keywords:** sustainable stormwater management; urban planning; institutions; institutional change

---

## 1. Introduction

The appropriateness of conventional stormwater management relying on piped drainage systems is being increasingly questioned in the light of critical sustainability challenges. Impacts from cloudbursts, excessive water flows and pollution, are already causing severe damage in built environments and receiving water bodies. These problems are expected to be further intensified due to shifting rainfall patterns related to climate change, aging infrastructures, and the continuous densification of cities, where impervious surfaces are increasingly replacing permeable surfaces and natural drainage [1–5]. In the Nordic region (Scandinavia/Fennoscandia), signs of this trend can be seen in the form of increased costs for flood-related claim payments by insurance companies [6] and difficulties meeting environmental quality standards for surface waters.

There is an emerging consensus that a larger proportion of stormwater should be managed above ground, via alternative control measures designed to mimic the natural functions of pre-development hydrology [7]. Such sustainable stormwater strategies aim to minimize impervious cover by promoting infiltration, ponding and the harvesting of rainwater. In this decentralized management approach, stormwater runoff and pollution are primarily controlled by measures located near the source to strive towards well-integrated measures that perform multiple functions, including flood protection, pollution removal and groundwater recharge, as well as recreation, biodiversity and urban aesthetics.

Hence, in addition to practical challenges related to design and construction, larger-scale implementation of green and blue solutions necessitates reframing stormwater management from a

technical problem handled by engineers at water utilities, to a broader sustainability issue more closely integrated in wider urban planning practice [8].

In the same vein, it has long been argued that the greatest challenge to transform the stormwater management sector, beyond solutions based upon piped networks, is not about advancing technology, but about developing new working procedures and planning routines that involve wider actor collaborations, in order to make stormwater a concern for a much more diverse and inclusive urban planning community [9].

While there seems to be a general agreement on the need for institutional change and improved collaboration in the sector, how to move forward remains elusive. The slow pace of the largescale and widespread implantation of sustainable stormwater management is a testimony to this situation. Previous research provides the foundation for understanding the main barriers to improved sectoral collaboration [10]. However, there is still a strong need for knowledge about ways to overcome these barriers, including the design of purposive and effective planning processes, as well as an improved understanding of how to support and enhance cooperative initiatives [7–9].

Based upon these initial assumptions, this paper aims to scrutinize the challenges pointed to as barriers for a transition to sustainable stormwater management (This article uses the concepts "sustainable stormwater management" and "alternative stormwater management" interchangeably when referring to non-piped systems that use nature-based solutions to manage stormwater, such as green and blue infrastructure. In the literature, this is sometimes referred to as "non-conventional stormwater management", as opposed to systems that use traditional underground pipes.) in Swedish municipalities with special focus on the interplay between formal and informal institutions in the sector. In a second step, and based on the investigation of the current challenges, we identify possible ways forward for municipal stormwater policy and planning. Three broad research questions guide our analysis:

1.  What are, according to central stakeholders, the major challenges to sustainable stormwater planning in Sweden?
2.  What changes are needed to enhance the development towards sustainable stormwater management in Swedish cities?
3.  What is the role of formal and informal institutions for understanding current challenges and ways forward?

Although the empirical materials for the study emanate from a Swedish municipal context, the overall analysis and study outcomes should be of interest to a broader audience of researchers and decision-makers beyond the Swedish context, since the challenges discussed are of a general character, and feed into discussions on the politics of urban stormwater management in the previous literature.

## 2. Background and Theoretical Departure

This section begins with a short introduction to the Swedish stormwater governance context. In the subsequent section we present the theoretical concepts associated with formal and informal institutions that have shaped our understanding of the dynamics and change in this sector. Finally, we turn to previous research on the barriers and challenges that organizations and actors face as they move towards new ways of thinking about and planning stormwater services in the urban context.

### 2.1. The Swedish Stormwater Governance Context

In Sweden, urban planning and stormwater management are municipal responsibilities. On a general level, the Swedish water sector has been influenced by international trends of the reregulation of utility markets, which has been ongoing since the mid-1980s [11].

This has implied the introduction of business management principles, including contractual relations and competitiveness, into a policy domain, which historically has been governed by public utility traditions and norms. However, even if many Swedish water utilities are currently organized

according to business management principles, they remain municipally owned. In this reformed institutional structure, stormwater management is usually the responsibility of the same utility company that manages drinking water and wastewater systems.

Swedish stormwater planning is further influenced by several domestic legislative frameworks, including the Planning and Building Act, The Swedish Environmental Code, and the Water Services Act. In some respects, these national legislations are considered ambiguous, which causes problems with respect to stormwater planning. For example, the Swedish Planning and Building Act has proven to be a weak instrument for regulating stormwater management needs in detailed plans [12]. Moreover, although municipalities are required to highlight, for example, flood risks in their comprehensive plans, these plans are not legally binding, making it hard to determine whether the risks are sufficiently managed.

The European Water Framework Directive (2000/60/EC) provides additional contextual background for water management in Sweden. Since 2004, this framework has been gradually introduced into the Swedish water administration, with the main objective of minimizing water pollution, as well as restoring the chemical and ecological status of streams and other bodies of water. This framework has institutionalized systematic work for monitoring environmental standards in the Swedish waters according to a pre-determined time cycle, and has introduced a new approach for managing and protecting water, based on the spatial extent of watersheds, rather than administrative boundaries. Despite these stricter regulations, most stormwater in Sweden is still untreated when discharged from built environments into surface waters.

As discussed above, power and control in the Swedish stormwater sector has gradually moved away from the traditional utility model toward supranational organizations, such as the European Union (EU). Moreover, prevailing management ideals promote nontraditional actor collaborations involving stakeholders with additional interests and issues. The latter refer to both utility companies and other municipal departments (e.g., urban planners, ecologists and landscape architects) that become stakeholders as stormwater measures develop into multifunctional solutions. Additionally, there is a strive towards public participation in order to increase the involvement of communities and citizens in taking responsibility for reducing stormwater runoff, by locating measures on private properties.

The next section looks more closely into the concepts of formal and informal institutions, as well as the process of institutional change that underpins our understanding of societal change and the contemporary challenges facing the Swedish stormwater sector.

*2.2. Formal and Informal Institutions, Path-Dependencies, and Institutional Change*

To analyze the cultures and processes of planning and decision-making, this study directs attention to the concept of institutions, here defined as the rules of the game that steer human interaction and provide the incentives structures for societal change and continuity [13]. Hence, institutions are normative by nature. Institutions shape the way we view the world and what we consider as rational solutions to problems. From this it follows that the meaning of rationality can differ both in time and space [14].

At the heart of any institutional analysis is the question of how institutions consciously or unconsciously impact the behavior of individuals or groups [15]. Institutions, as defined here, can be either formal (e.g., law, policy and regulations) or informal (e.g., conventions, norms and traditions). Whereas formal institutions can change overnight, informal institutions change more gradually. Importantly, institutions as defined here should not be confused with organizations, which are actors or players within a certain playing field [13].

Following from the above definition, institutions in the context of this article influence the action and thinking of those involved in or excluded from formal planning procedures on sustainable stormwater management.

Institutions not only shape the way that we look at problems, but also are the products of how we perceive an issue. If formal institutions change, informal institutions (e.g., social norms, practices

and informal routines) are likely to change as well. This process can also work from below with new ideas, norms and evolving practices, enforcing or inspiring formal institutional change. Moreover, the path-dependent nature of institutions may impede action and cause a reluctance to change [13]. This, as argued by Healy, typically happens as norms and procedures from a previous governance regime become embedded in the practices of the next [16].

Although not always explicitly discussed in the same theoretical terms, much previous research has directly or indirectly addressed the formal and informal institutional aspects of stormwater planning and decision-making, including how they interact and how institutional factors influence actor collaboration. The following section illuminates some conclusions drawn by the previous literature, with focus on the barriers and challenges that organizations face as they move towards new ways of thinking of and planning stormwater services within the urban context.

### 2.3. Current Challenges to Sustainable Stormwater Planning

Previous research on sustainable stormwater management and planning has pointed to the importance of institutional factors to understand and address current challenges. In addition to geophysical constraints relating to, for example, climate, hydrology, land, soils and topography, "law and social factors", are presented as imperative for sector dynamics [7].

Many researchers have highlighted the need for more integration and cross-sectorial collaboration as being key to the transition from conventional to sustainable stormwater management. For example, Serrao-Neumann et al. [9] note that water resources management in Australia is undertaken by multiple agencies, without being coordinated and/or integrated to address the total water cycle: "Urban systems are not isolated entities but exist within a landscape and hydrological system ( . . . ) beyond jurisdictional boundaries". To address environmental and hydrological connections between cities and their regions, these researchers argue for a better integration between land-use planning (both urban and regional) and water resources management.

Similarly, Dunn et al. [17] found that the current compartmentalization of urban water services into water supply, wastewater collection and stormwater drainage, is a typical product of a rationalistic scientific paradigm focused upon controlling, predicting and planning (i.e., command and control engineering), as opposed to a systems-thinking approach that embraces the complexity and unpredictability of natural resources management. In the same vein, Fergusson et al. [18] show how a central characteristic of creating an enabling institutional context for sustainable stormwater management in Melbourne, Australia, was to give up the idea of control and predictability. Instead, they argue that systems and structures must be designed based on the idea of adaptability and flexibility in order to meet unanticipated severity of cumulative impacts of climate change and natural rainfall variability. This, for example, is illustrated by a move away from a reliance upon large-scale, centralized infrastructure, where "water professionals worked from the assumption that environmental variation could be predicted or controlled through technical solutions based on historic data" [18], towards systems based on more decentralized and complementary measures that encourage source control rather than merely treating the symptoms [19].

One way of triggering systems thinking and perspectives is to establish sector-wide actor collaborations early in the planning process, as well as to identify shared goals for common planning [8]. In this context, the lack of informal institutions, such as a culture of cooperation, a lack of trust between different departments, and the great need for social capital (i.e., trust, social networks and norms of reciprocity), can be barriers to system change [10]. In addition, the lack of a supportive legal framework and the political mandate to work towards alternative stormwater management solutions, is discussed as a factor that causes a resistance to change [7]. For example, fragmented or unclear responsibilities often impede progress, since several municipal entities are expected to collaborate on multifunctional solutions and at watershed/landscape levels.

While focusing on low impact development (LID) practices in a stormwater management context, Kim et al. [20] conclude that a lack of incentives or policy instruments and clients' lack of knowledge

and the lack of a "development team" are major bottlenecks to the implementation of alternative stormwater measures. Education programs and innovations in policy systems and regulations are suggested to be the most successful interventions for elevating proper LID practices.

Others have emphasized the sociotechnical character of system change and continuity as ways to understand sector dynamics. To explain how social and technical choices are shaped by the institutional environment, Goulden et al. [21] draw upon Scott's [22] three-pillared, socio-institutional framework: Cultural–Cognitive (frames and discourse), normative (outspoken policy, goals and professional norms), and formal laws and regulations. Similarly, Fergusson et al. [18] use the same theoretical framework to analyze how to create enabling conditions for integrated water management in Melbourne, Australia. Here, changes took place at three interactive levels. These included a cultural–cognitive shift among water professionals from predictability and control to adaptability and contingency planning, as well as new knowledge through experimentation, demonstration, evidence and sector-wide learning. A normative shift took place through a growing emphasis on waterway health, which triggered an increased focus on stormwater quality treatment, wastewater recycling services, and policy on sustainable water management more generally. Strong government leadership and public support similarly reflected the major cognitive and normative shifts, which contributed to an acceptance and a commitment to alternative management strategies and water saving measures. This was followed by regulative changes towards cooperative institutional mechanisms, which contributed to collective approaches, rather than administrative silos. Regulative changes also included strong economic incentives for market actors to invest in sustainable solutions and government subsidies for innovation and water efficiency measures [18].

Cettner et al. [23] take a long-term, historical perspective and look more closely into the socio-institutional barriers that are decisive for system change. The authors point to a pipe-bound mentality and system culture as a major obstacle causing resistance to change in this sector. The challenge then lies in breaking away from the technical, but even more so the institutional, path-dependencies, by framing stormwater planning as something "more than a pipe-issue". Here, they show how stormwater management is seen as a technical problem to be solved, rather than an opportunity for improvement or a sustainability issue. Besides the role of institutions, Henestra et al. [24] point to ideas, actors and interests as similarly important to systems change. In our study however, a conscious choice has been made to focus on the role of institutions as the "the rules of the game" that seemingly steer what actors will perceive as their interest.

In summary, the use of piped drainage as a model for urban development is increasingly being questioned, and some of the challenges for a transition to sustainable stormwater management have been identified. Many, researchers as well as practitioners, agree upon the need for institutional change; however, less is known about exactly how to move forward, what steps that need to be taken in different contexts, and why. Empirically situated in a Swedish municipal context, and building upon previous literature, this study moves beyond the description of challenges to suggest ways forward that could enhance the integration of sustainable stormwater management in urban planning. This task requires a closer investigation that unpacks the central challenges generally described in the previous research in more detail, in order to gain a deeper understanding of the changes needed.

## 3. Materials and Methods

This study is based on data collected through a series of workshops, individual interviews, and a questionnaire targeting stakeholders in the Swedish stormwater sector. The data collection aimed at building an understanding of the current planning challenges and changes seen as necessary by practitioners in the field.

Seven stakeholder workshops were held with a broad group of stakeholders from different sectors (Table 1). Held at the Linköping University Campus in Norrköping, workshops 1, 2, 7 and 8 were attended by actors from across the country.

**Table 1.** Overview of stakeholder workshops and professions of the participants.

| No. | Location | Date | Participating Stakeholders |
|---|---|---|---|
| 1 | Norrköping | 8 February 2017 | 13 (6 planners, 2 water utility representatives, 2 government officials, 1 consultant, 1 branch association representative, 1 constructor) |
| 2 | Norrköping | 17 November 2017 | 16 (6 planners, 2 water utility representatives, 2 consultants, 2 government officials, 2 branch association representatives, 1 constructor, 1 visualization platform developer) |
| 3 | Norrköping | 7 May 2018 | 3 (all planners) |
| 4 | Linköping | 22 May 2018 | 3 (2 planners, 1 water utility representative) |
| 5 | Malmö | 31 May 2018 | 4 (3 planners, 1 consultant) |
| 6 | Göteborg | 11 June 2018 | 4 (2 water utility representatives, 2 planners) |
| 7 | Norrköping | 2 May 2019 | 7 (2 planners, 2 government officials, 2 consultants, 1 branch association representative) |
| 8 | Norrköping | 3 June 2019 | 10 (3 water utility representatives, 2 planners, 2 consultants, 2 branch association representatives, 1 property developer) |

Workshops 3–6 were held with participants from specific municipalities who had extensive experiences in collaborating across sectors in stormwater planning.

The workshops followed the form of a stakeholder dialog, described as "a structured communicative process of linking scientists with selected actors that are relevant for the research at hand", aimed to "link different domains of discourse" or different types of knowledge and ways of knowing [25]. This method can deepen researchers' understanding of an issue or context, and provide a useful reality check on the results gathered [26]. Topics addressed in the workshops covered challenges to current stormwater planning procedures, decision support needed to advance planning, ecosystem-based planning, and cross-sectoral collaboration and dialog. Since a series of workshops have been held within the project, this offered opportunities for progression along the way, where dominant themes and central challenges were identified at an early stage, and then the issues were dissected even more deeply on the next occasions. We always tried to move beyond discussing challenges by also considering potential solutions on how critical issues could be dealt with, and how needs could be met.

In addition to the discussions held at the larger workshops, 12 semi-structured, individual interviews with municipal planners and sector professionals were also conducted (Table 2). The interviews, which were carried out in 2017 and 2018, lasted between 30–60 min, and offered more in-depth insights on specific issues or processes in the stormwater management sector. These interviews also provided an opportunity to capture views that might not always be voiced in larger gatherings. The interviews focused on the barriers and opportunities for sustainable stormwater management, with special focus on institutional challenges and needed improvements. In one of the municipalities, the interviews focused on following the vertical planning process, so several respondents throughout the implementation chain were interviewed. Since more interviews were carried out in this specific municipality, this could potentially create a bias. Our assessment, however, is that the topics discussed have been raised several times in other forums, including our workshops, so that the results from these interviews should be valid in a wider context.

**Table 2.** Individual Interviews carried out for the study.

| No. | Municipality Code | Affiliation |
| --- | --- | --- |
| 1 | A | Head of Department, Planning Department |
| 2 | B | Water Planner, City Building office |
| 3 | B | Water Utility Officer |
| 4 | C | Project Leader, Public Works, City Building Office |
| 5 | D | Environmental Strategist, City Building Office |
| 6 | E | Planning Department, Water Utility |
| 7 | F | Stormwater Coordinator, Environmental Office/Land and Exploitation Office |
| 8 | F | Project Leader/Land- and Exploitation Engineer Planning Office |
| 9 | F | Project Coordinator, Water Utility |
| 10 | F | Operation and Maintenance, Engineer, Office of the Urban Environment |
| 11 | F | Comprehensive Planner, Planning Office |
| 12 | F | Building Inspector, Building Permit Office |

The interviews and workshop discussions were audio-recorded and transcribed word by word. Transcriptions were inductively analyzed with the broad aim and research questions in mind. Themes concerning institutional challenges and needs were highlighted and thematically categorized according to the recurring sub-themes and patterns that emerged from the analysis. The rich and extensive materials also allowed for cross-checking between different data sources.

The survey was designed to reach a broad group of planners, decision-makers and water system practitioners. Questionnaires were therefore distributed at two major national conferences (*Vatten, Avlopp, Kretslopp* (Water, Drainage, Re-circulation) held in Norrköping 16–18 March 2017 and *Rörnät och Klimat* (Drainage and Climate) held in Malmö 29–30 March 2017) on sustainable water management. A total of 116 respondents answered the questionnaire, including 60 water utility officials, 25 municipal planners or officials, 20 consultants, 10 researchers or "other", and one politician. The survey has primarily been used to crosscheck if any of the institutional changes that were seen as necessary by respondents in the workshops and individual interviews are consistent with the perceptions among a broader group of water professionals; particularly water utility representatives who were generally underrepresented in the workshops and interviews. Consequently, only the responses on two particular questions have been incorporated into the results. For these, respondents were asked to rank which plan processes they see as most important to develop, followed by a free-text question asking how municipal planning can best be developed to spur sustainable stormwater solutions.

In addition to engaging in the workshops held within this specific project, the researchers also participated in several conferences, consultancies and adjacent research projects on urban climate adaptation more generally, including stormwater management. This has allowed for an interaction with central stakeholders outside the context of this research project. Experiences and background knowledge provided from adjacent contexts are not referred to as empirical materials in this study, but have informed the direction of the study, including the formulation of the aim and research questions.

## 4. Results and Analysis

This section presents and analyzes the governance challenges and needs for improvements in Swedish municipal stormwater planning, as portrayed in individual interviews, workshop discussions and survey data. Based upon this analysis, we continue to discuss enablers for sustainable stormwater planning in Sweden.

Special attention is paid to the role of formal and informal institutions for understanding current sector dynamics, as well as ways forward towards a more enabling institutional context. An overview of the results from this section is attached in the Electronic Supplementary Materials, see Table S1.

*4.1. Current Challenges in Municipal Stormwater Planning*

4.1.1. Putting Stormwater on the Agenda

A city typically can be viewed as an arena where the battle between interests is materialized in the urban space. That is, city environments in many ways mirror the preferences of those in a position to influence the urban agenda. Similarly, spatial requirements related to flood control and stormwater management are being constantly, implicitly or explicitly, negotiated [16,19].

Several actors in our study argue that in the eyes of politicians and centrally positioned planners, the stormwater issue is sometimes considered a hurdle to urban development. To a large extent, this attitude is generated by a massive pressure on finding new land for property development in expanding large and mid-sized Swedish cities. According to the stormwater coordinator in Municipality F, the EU Water Framework Directive has proven to be a very weak instrument for claiming the spatial needs for stormwater in urban development. Some respondents also testified that raising the stormwater issue in the planning process many times brings an atmosphere or feeling of inconvenience, as if one "messed up" or disturbed the decision-making process (Interviews 1, 2 and 3).

Our analysis shows that goal conflicts are not always explicitly identified in plans, but remain unsolved and embedded in strategic land-use decisions. That is, by overlooking the spatial requirements of flood and pollution control, stormwater needs are being efficiently marginalized from political agendas, and in the end, land is not reserved for this purpose in the physical environment. As firmly stated by a municipal landscape architect:

"We will get nowhere if we are not allowed to set the stormwater issue against other issues" (Workshop 4).

In this context, we see that a visualization of goal conflicts would be a way of transparently revealing how priorities between competing interests are made. This in turn is a way of avoiding the built-in and inherent conflicts between misaligned strategic goals, conflicts that instead will be passed to other actors to deal with in the later stages of planning.

Our empirical material shows that land-use decisions in comprehensive planning are frequently made on questionable grounds without enough consideration as to how risks related to floods and pollution are to be handled in those later stages of planning. These challenges are then handed over to water utilities with the simple instruction to "solve the stormwater problem" (Workshop 4) within the boundaries of individual detailed plans. However, several respondents argue that the preconditions for sustainable solutions within such small areas are not favorable, since strategic land-use decisions have already been taken, and there is not enough land available to take care of the required volumes and flows of stormwater (Workshops 3, 4 and 5). Therefore, we argue that issues related to runoff and pollution control, as well as to the integration of water in the urban environment, are more efficiently handled in wider geographical areas.

Along the same lines, workshop discussions show that it is important to more explicitly address water-related risks and needs from a larger spatial scale (i.e., the landscape/ecosystem perspective in the comprehensive planning phase). Since comprehensive plans involve a wider area, this offers opportunities for planning stormwater management that consider water flows and movement on the watershed scale. To do so, respondents considered it as central that planning is based on more extensive investigations of watershed scale preconditions for stormwater management, including current and future water flows in the landscape, and how urban densification affects flood risks and water pollution (Workshop 1). Here, taking a watershed perspective was seen as a key, not only for identifying goal conflicts, but also for developing strategies to solve these conflicts.

If building such discussions on common knowledge grounds, such as stormwater assessments based on watershed scale, multifunctional solutions designed according to the local context were also seen as easier to motivate, and maladaptive measures easier to avoid (Workshop 3). An important challenge, however, is to get this message to politicians and comprehensive planners in municipalities who lack experience with flooding or pollution. Several workshop participants argued that experts in

stormwater management must be better at specifying the effects of various land-use options and what it takes to manage risks associated with flooding and pollution, rather than only stating risks. These participants believed that this approach would lead to constructive dialogs:

"What we need to show are the benefits. What happens if we do not take any measures? What happens if we do? What effects do these have? Maybe it does not solve all the problems, but such overviews would help ( ... ). We do a lot within detailed plans, but not in the bigger context (on a watershed level). In the end, the most effective measure may simply be not to build in unsuitable locations, but this is not considered." (Workshop 6)

Related to the above, respondents emphasized the need to make such information more accessible to local politicians. A stormwater coordinator, for example, noted that if the locations of and specifications for stormwater management were identified on maps, it would be easier to know how much surface water and how much purification is needed within each detailed plan, and how much land is needed for these purposes. Such maps were also seen as an important knowledge foundation and a basis for joint planning between different municipal entities (Workshop 6).

Survey data point in similar directions. When asked how municipal planning can best be developed to spur sustainable stormwater solutions (free text question; n = 38), the respondents most frequently raised the need for including stormwater management considerations already in comprehensive planning, obligating stormwater investigations before making plans, and enhancing common learning about stormwater risks and solutions. Comprehensive planning was further seen as the most important plan process to develop.

To sum up, our results suggest that the needs and risks related to stormwater must be raised more explicitly in comprehensive planning, and that this requires multi-stakeholder dialogs for common learning processes. In this context, institutionalizing a view of stormwater as an issue to be analyzed and planned at a wider watershed level would be a way of visualizing goal conflicts, and thus raise stormwater concerns on planning agendas. This illustrates how considering stormwater as part of an ecosystem, rather than a technical issue to be solved within a single detailed plan, also brings about significant changes to the institutional environment.

### 4.1.2. "We Are Co-Parts, Not Counterparts": The Challenge of Establishing Common Visions and Overcoming Silos in Stormwater Management

The empirical evidence from this study confirms conclusions from previous research. Specifically, a lack of a collaborative culture is considered a major bottleneck to the implementation of sustainable stormwater management in Sweden. This bottleneck, we argue, in many ways is generated by current, historical and path-dependent institutional arrangements governing the sector. When pipe-bound stormwater infrastructure was the prevailing technical choice, engineering expertise on dimensions and expected flows were obviously central. As of today, a new way of handling water in the urban space generates a need for additional competencies and new organizational arrangements.

Managing stormwater above ground often means that the measures have more than one function, with several actors and agencies being involved in the design, operation and maintenance. Yet, cocreation requires leadership. From an organizational viewpoint, the issue of alternative stormwater management has no clear institutional affiliation. That is, several sectors might have a stake, but none hold a position of leadership that can coordinate the procedure or make decisions affecting others. Such leadership, however, is seen as crucial for moving forward:

"All sectors are equal in terms of negotiation power but have different mandates and responsibilities. You have your area but are not able to influence any other sector areas. Then a superior level is needed—position or something that is able to coordinate or steer or something." (Workshop 3)

As mentioned above, the lack of a coordinating function causes inertia and inhibits the working process. This condition creates the risk that business-as-usual procedures and solutions are maintained: "The fastest way is just to do as usual" (Workshop 4). Therefore, to break path-dependencies in current working procedures, formal institutions need to change so that they foster cocreation mentalities. What

is needed here is leadership with a mandate and capacity to mobilize actors towards a common goal. Another requirement for moving forward is that responsibilities are made explicit for the different aspects of stormwater management. This again necessitates that leaders at a strategic level actively engage in making decisions about roles and mandates, as well as how the division of responsibilities is made clearer.

As highlighted in previous research [8,10,18], a major constraint to cocreation in planning is the frequently discussed topic of silo-thinking. This phenomenon is partly generated by the sector-oriented character of the organization and includes the problem of silo budgets. Such silo budgets seriously reduce efficiency by putting the wrong incentives in place, and this thereby fails to generate the much-needed cooperation between departments:

"If the problem at hand is of a silo character, then I am totally ok with the fact that working procedures and solutions are so as well. But if the problem is boundary-spanning/cross-sectorial, then solutions must also be of the same character." (Workshop 6, water utility representative)

Working in silo structures, it was argued, generates unnecessary questions regarding who is responsible for construction, operation and maintenance. These problems are particularly pronounced in the case of multifunctional stormwater solutions where responsibilities are easily blurred. Traditionally, water utilities take care of stormwater measures. However, since open and nature-based solutions have several purposes, the division of responsibilities between actors is easily blurred:

"Somewhere there is a limit as for how much that can be paid by the water and sewage tax-payers. I mean, they cannot carry the cost of a park so to say." (Workshop 4, water utility representative)

This is but one illustration of a situation where new technical solutions entail that the old institutional arrangements are no longer suited for the new circumstances.

An additional challenge relates to prevailing management ideals, which implies that actors to a large extent are divided by roles, such as clients and contractors (inspectors and performers), rather than as co-producers of sustainable solutions. This model is not perceived as beneficial for pursuing collective purposes and forming a culture of cocreation, for example, in situations where plans are dismissed by the county board or the environmental office. Planning office actors argue that it would have been more productive to have a dialog around critical issues when plans are designed. As of now, time is invested in working on solutions that will not be accepted:

"So, in fact we do want the same thing, we are not counterparts but co-parts ( … ) however, that kind of cocreation culture is not particularly strong." (Interview 4, stakeholder from Municipality C).

This institutional arrangement, we argue, also hampers the development of trustful relations, which are needed for cocreation processes and generate a culture of referral rather than a co-development of plans.

Moreover, water utility representatives are frequently treated as consultation bodies in holistic planning, rather than as co-planners. When involving water experts after the initial land-use planning, one runs the risk that this results in an uncooperative attitude from these actors. Stakeholders noted that a natural reaction is to argue for why various areas are unsuitable for development. A more constructive approach would be to discuss areas that are suitable for a specific type of development, and what measures are needed to ensure a resilient development (Workshop 3). That is to say, approaching central stakeholders as consultation bodies rather than as full members of a cocreation process seems to create distrust and seriously impede critical development processes.

This is another example of how institutions (i.e., the rules of the game) can affect working cultures and in the end influence the quality of process outcomes.

### 4.1.3. Vertical Collaboration—The Gap between Intentions of Plans and Implementation

Another critical issue frequently mentioned in individual interviews and workshop discussions is the gap between the intention of plans and the implemented realities. This raises questions about how to ensure that the visions and ideas from the comprehensive planning stage are integrated in detailed

plans, how to ensure that these planning ideals are passed on and realized by contractors and property developers, and ultimately how they are followed up by the building permit office.

Planners participating in this study experienced difficulties controlling the outcome of the plan instructions. The underlying cause to this gap is framed as a problem of transferring or handing-over responsibilities for a project between different stages in the planning process (Interview 7). Here, information, knowledge, and intentions are perceived as lost along the way from the formulation of a plan to the implementation and service delivery stage.

The same applies to the stage where plans are to be realized by property developers, builders and the operators of stormwater measures. In this context, planners view a lack of knowledge and awareness among practitioners as a barrier for fulfilling the plans. They argue that there may not be a thorough understanding of the reasons for why a certain solution was selected, a restriction was made, or a decision was taken. This, as argued, could explain why sustainable stormwater management ideals do not reach the implementation stage (Interviews 7, 8, 9, and 11). In addition, the lack of routines for monitoring and evaluating does mean that there is no real control over whether plans are followed (Interviews 7, 8, 9, and 11, Workshop 8). From a bottom-up perspective (i.e., property developers and permit officials), the problem is framed as a lack of consideration for feasibility during the planning process, as well as (again) a lack of knowledge about the on-the-ground realities among the centrally positioned planners:

"There are many different formulations regarding stormwater in the detailed plans. This is partly because my dear colleagues at the planning department do not have a very good understanding when it comes to stormwater issues; it is something they may have heard of but do not know very much about. And then you can write almost anything so to say." (Interview 12).

According to a stakeholder from municipality C, decisions on practical measures are sometimes made too early in the process, and by people who do not necessarily have the right competence to assess solutions in different contexts (Interview 4). Others, on the other hand, argue there might be either too little guidance or even contradicting orders in the directions attached with the plans received by the building permit office (Interview 12). These problems could be resolved by having a stronger involvement and dialog between actors during the initial stages of a planning process when the decisions are made. Here, communication is needed to support mutual learning and to foster trust and understanding between different parties in the process.

To sum up, our empirical analysis shows different, yet interrelated, major challenges to sustainable urban stormwater management. First, stormwater needs to be made more visible in order to be integrated into the planning. As of now, stormwater management is handled too late in the planning process. Therefore, plans do not consider water from a landscape/ecosystem perspective. Secondly, the stormwater issue has no clear affiliation or leadership. Being divided by several units, the sectoral division and budgets are not optimized for stormwater planning. Moreover, a culture of referral rather than cocreation between sectors and units cause serious impediments to the process. Third, the vertical planning process for stormwater management often has a weak continuity of ambitions between comprehensive planning, detail planning and building permission. Property developers and permit officials are not involved in the design of stormwater measures. Finally, there is no clear and consistent system in place for securing that stormwater intentions are realized.

*4.2. Ways Forward—How to Enhance Institutional Change and Collaborative Action in the Stormwater Sector*

As shown in the previous sections, many of the current challenges to Swedish stormwater management can be traced back to current institutional frameworks that are not supportive for new ways of governing this sector. Thus, responding to the challenges portrayed by our respondents, the following section depicts what we consider as a set of important steps towards a more enabling institutional environment. We will explicitly discuss how our suggestions would affect or alter the formal and informal rules of the game to foster long-term change towards sustainable stormwater management.

### 4.2.1. Making Space for Water—On Political Agendas and in the Landscape

As previously argued, the development of urban stormwater infrastructure must be recognized as more than a value-neutral and technical planning process. To understand sector challenges, we need to recognize the power and politics that are inherent and decisive for systems change. Here, institutions (the rules of the game) can contribute to shaping interests, and thus mobilize actors towards common goals.

As pointed out in Section 4.1.1, a transition to sustainable stormwater management requires that plans take a watershed perspective as a departure point and as a basis for further planning. Therefore, our first suggestion is to make an assessment of the risks, needs and benefits related to stormwater management at a watershed level a mandatory step of the planning process. Such an assessment would contribute to placing stormwater in the spotlight and provide a better basis and a starting point for negotiations on how to reserve space required for stormwater management as early as possible in the comprehensive planning phase. Making a watershed assessment compulsory would also be a way to force decision makers to adopt a landscape/ecosystem-based thinking in land-use decisions and spatial strategy making. This approach would raise the stormwater issue on political agendas and initiate a learning process where the spatial needs for flood and pollution control in the long-term are mainstreamed and integrated as a natural part of physical planning. In addition, it could prevent a situation where "storm water problems" are passed on to be handled by engineers within smaller residential areas spatially defined by detailed plans.

In this case, as we have seen, formal incentives must be put in place to foster a change in informal norms and practices, where the natural water flow and hydrological context are seen as an integral part of physical planning. The larger story here is about fostering a mindset of working with nature rather than taming nature [19]. Hence, in this context, it is key that stormwater issues are not seen as a problem for the engineers and water professionals to solve after all of the strategic and over-all land-use decisions have already been taken.

In this context, our stakeholders suggested that stormwater management should always consider several detailed plans for each watershed. In Swedish planning terminology, this can be made in what is referred to as a "plan program". As commonly used, a plan program is developed to review the conditions and visions in a larger area to provide a structure for buildings, road networks and green spaces for subsequent detailed plans (Workshop 5). However, this is not a standard procedure in municipalities. Institutionalizing the plan programs that specifically target stormwater was considered crucial for building better overviews of how much surface water can and should be treated within the various detailed plans, and how much densification is possible. It was also seen as an important basis for constructive dialogs between various planners and with politicians about potential land use (Workshop 5).

Moreover, making space for water would also imply changing established norms and practices on how borders for detailed plans are being drawn. A way forward here is to institutionalize routines where the planning of stormwater is made for areas in the urban landscape, rather than for areas within detailed plans. So far, the built environment has been the decisive factor for how detailed plans are being delineated in the spatial planning process in Sweden.

Another important way forward is to similarly reserve space in the detailed plans for nature-based solutions aimed at managing stormwater above ground and thus changing the normative practice for how plans are drawn. In this way, reserving land in the detailed plan map would be a consequence of having the issue recognized on the political agendas of urban planning. However, changing formal procedures could, in the long term, foster a new way of thinking about water in the urban landscape. The next step, of course, is the actual physical manifestation of planning ideals and having the plans realized in the construction process. This relates directly to the monitoring of the planning process, which will be discussed in Section 4.2.3.

4.2.2. Designing an Organization for Sustainable Stormwater Planning

When underground pipes are the dominant technical choice, engineering expertise on dimensioning and flows is central for stormwater management. Today, as we have seen, a new way of handling water in the urban space generates an increasing need for new competencies, as well as a restructuring of the internal organization. For example, taking care of stormwater above ground means there is not only a need for experts calculating the dimensioning of pipes, but also for project leaders and planners with skills to develop multifunctional spaces in addition to the maintenance of open solutions. Whereas conventional stormwater management has been organized according to silo principles with a heavy emphasis upon traditional engineering perspectives, multifunctional solutions require cocreation between different departments and entities. This implies a new role for water engineers, and a need for water utilities to build in-house planning and project leading competence to coordinate projects involving a more diverse set of actors.

The empirical evidence from this study clearly confirms conclusions from previous research that shows that a lack of a collaborative culture is a major bottleneck to the implementation of sustainable stormwater management [8,23]. This, in many ways, is generated by current, and to some extent historically path-dependent, institutional arrangements governing the sector. Ways forward here imply finding the means to institutionalize the cross-sectoral coordination of stormwater planning. In the following, we provide a few suggestions for how to change the formal and informal rules of the game in this direction.

First, as discussed above, cross-sectorial coordination will never be realized without a coordinating or convening function designed to mobilize actors towards a common goal. This means that a leadership function is needed with a mandate to make decisions that affect other sectors and actors. Based on this and our previous research in the field [12], we suggest that stormwater should be centrally coordinated by transferring leadership and coordination to a central municipal management office, which often has a more direct influence on municipal sector boards and their budget processes.

Second, such a new way of coordinating stormwater management and planning should be followed by financing models that release funding for cross-sectoral projects and measures. Whereas a traditional sector division would demand funding from all of the sector boards, we suggest that resources for urban development should be pooled to enable funding for cross-sectoral projects and measures. This would mean distributing less money to sectoral boards and investing in joint initiatives. This approach has been tested successfully in a few Swedish cities [12], and could be used as a model for other municipalities.

Third, as pointed out by Fergusson et al. [18], the artificial division of urban water into separate and sometimes incoherent strategies for drinking, stormwater and wastewater, results in inconsistencies and misaligned strategic goals. For example, plans for the location of stormwater flooding surfaces need to consider downstream watersheds for drinking water, the planning of drinking water intakes, and possibly the location of waterworks. Clearly, much would be gained from integrating various plans and policies regulating urban water management in order to find long-term and sustainable solutions to different but interconnected challenges.

4.2.3. Ensuring Continuation and Implementation of Stormwater Ambitions

As argued in Section 4.1.3, the gap between policy and implementation can be understood in different ways. The "top-down" perspective, as we have seen, suggests that information is lost along the way from the formulation of plans to the implementation stage due to a lack of top-down control. This way of framing the problem implies that a better control of the process would secure better outcomes. The bottom-up perspective, on the other hand, suggests that planners lack knowledge about the operability and preconditions for implementation. We argue that both perspectives need to be recognized, and that a combination of solutions would move us forward.

To begin with, as previously discussed, the cocreation of plans is also crucial between the different stages and actors in the vertical planning process, including the building permit office. This is

important in order to build an understanding of why certain decisions were made, and to openly discuss priorities and the reasons behind different strategies or standpoints. A solution here would be more interaction, collaboration and participation during the plan formulation process. Networks or forums for knowledge transfer and common dialog that foster learning and mutual understanding between the different parties in the implementation chain are important here, but also to improve the routines for ensuring that experiences and knowledge from "below" are brought back into the process. To ensure that any lessons learned do not just stay with practitioners in the operation and the maintenance stage, our stakeholders suggest that feedback should be made an institutional routine (Interview 7). In some municipalities, formal networks and discussion forums are already used, but more importantly the people who attend these meetings should have the knowledge and courage to voice their concerns. This again depends upon the importance given to sustainable stormwater management in overall strategic goals for municipalities, and whether or not the issue is recognized at a political level. This approach would provide leverage for arguments, as they would be backed up by a formal political mandate.

Moreover, informal dialog and interaction are both crucial for building the trust needed for enhancing cooperation and building understanding between the different parties in the vertical planning process. Therefore, an important factor that enhances cooperation and dialog is to physically place the offices of representatives from different planning stages in the same building or corridor, so as to enable easier and more informal interaction. This arrangement would directly contribute to fostering trustful relations, as people would more easily be able to meet and communicate face-to face on an everyday basis. Facilitating informal interactions could be the most important way of building the cocreation cultures which are so much needed to enhance the integration of sustainable stormwater management in urban planning.

Our stakeholders suggested that the original author of a plan should follow the process all the way from the comprehensive planning stage to building, in order to secure that the plan's intentions are fulfilled. As previously mentioned, those people responsible for implementing any solutions are not necessarily aware of why plans come with certain restrictions, or why certain decisions were made in a previous stage. Having one person follow the whole process ensures the persistence and continuity of stormwater plans.

Finally, as discussed in Section 4.1.3, there is a lack of clear and consistent routines for following up on whether plans are followed. Therefore, a consistent monitoring system to follow up plans is needed, together with a clear mandate and the resources to do so. Institutionalizing such a procedure unmistakably signals that the issue is prioritized from a political level.

## 5. Discussion and Conclusions

This paper set out to scrutinize the challenges to and the needs for improving sustainable stormwater planning and management in Swedish municipalities, with special focus on the interplay between formal and informal institutions in the sector. Where previous research largely has stayed with describing the challenges to the implementation of sustainable stormwater management, this study explicitly strives to move beyond the barriers to also discuss the enablers and the ways forward in a Swedish municipal context.

The study has demonstrated the importance of the formal and informal rules of the game as a decisive factor to the dynamics in Swedish municipal stormwater planning. We know that reluctance to change and a resistance to altering routine procedures might be inherent in many organizations. However, even when there is a willingness to change, we observed how formal institutions in many ways inhibit the development of sustainable solutions. Formal and informal routines and practices, such as sectoral divisions and budgets, as well as client–contractor relations, discourage trustful relations and sometimes divide actors rather than unite them towards common goals. In our case, we also observed how path-dependent, institutional patterns cause a resistance to change. As discussed in the writings above, this can be exemplified by a lingering view of stormwater as a technical and

site-specific challenge to be solved by engineers within the boundaries of single detailed plans, whereas green–blue solutions rather require a landscape and ecosystem approach. Therefore, to formally institutionalize such a watershed analysis would be a way of inducing another approach in looking at water in the urban environment; i.e., to change informal norms (institutions) and in the end effect the way that stormwater is handled in the urban planning process.

We argue that to truly understand and address the challenges for a transition to sustainable stormwater, we need to address the complexity of both the formal and informal rules of the game, as well as how they interact.

The suggestions brought forward in this paper therefore consider the interplay of formal and informal institutions that are decisive for breaking with old development patterns that are not fully compatible with new goals. Suggestions are directed to a broader group of policy makers and planners, as well as researchers actively engaged in topics related to urban planning and sustainable stormwater management.

Our analysis has identified three different, yet interlinked, problem areas that stand out as of special importance for understanding the present state of and ways forward for urban stormwater planning in Sweden. A first challenge is that, from a traditional planning perspective, stormwater management is still to a large extent considered a site-specific technical issue to be handled by engineers and water professionals within single residential areas following the spatial boundaries of detail plans. This framing depoliticizes the issue of stormwater management to the extent that it is not made visible enough when planning for strategic land-use decisions using a larger landscape perspective. A second challenge is that current working procedures constrain the much-needed development of cross-sectoral working procedures and that they also put incentives in place that fail to foster cocreation cultures and mindsets. When striving towards alternative solutions, responsibilities and mandates are blurred, because the issue has no clear ownership or institutional affiliation, and because budgets are restricted to sectoral investments. In addition, actors are often divided according to their roles as clients and contractors, rather than combined as cocreators of sustainable urban environments. Third, there is a great need to ensure that planning ambitions and ideals regarding sustainable stormwater management are maintained throughout the different stages of the planning process, as well as in construction practices and in land use. This is what we refer to as vertical collaboration in this paper.

In response to the challenges pointed out in the above, we suggest several ways forward, summarized in the words below:

- Assessments of stormwater risks, needs and benefits at a watershed level should be made a mandatory step and a basis for comprehensive and detailed planning. We argue that this will make stormwater concerns more visible in negotiations on strategic land-use decisions, and therefore more easily integrated in urban planning.
- Leadership, as well as funding, for joint sustainable stormwater solutions should be transferred from sectoral boards to a central municipal office, which has a mandate to put actors on track towards common goals and distribute funds for earmarked projects.
- Plans regarding drinking water, sewage and stormwater should be integrated to avoid misaligned strategic goals.
- To ensure that stormwater ambitions are maintained from the planning to the implementation stage, networks for vertical interaction that facilitate mutual learning and dialog must be put in place.
- Routines for following up the intention of plans must be institutionalized, together with a mandate and resources for those in charge. This practice would also give unmistakable signals that the issue is being prioritized from a political level.

We argue that institutional changes in line with the points raised above will facilitate planning towards more sustainable stormwater management in Sweden and beyond.

**Supplementary Materials:** The following are available online at http://www.mdpi.com/2073-4441/12/1/203/s1, Table S1: Overview of perceived challenges for and needs to improve sustainable stormwater planning and management in Sweden, and proposed ways forward through institutional developments.

**Author Contributions:** A.B., E.G., and M.K. have been engaged in organizing the workshops and empirical data collection. A.B. and E.G. designed the study and analyzed the materials. A.B. coordinated the study. A.B., E.G., and M.K. contributed to the writing process. All authors have read and agreed to the published version of the manuscript.

**Funding:** This research was funded by Formas, grant number 2016-20090 and The Swedish Water and Wastewater Association, grant number 16-117.

**Acknowledgments:** We are grateful to all those individuals who generously shared their time to participate in the workshops, interviews and surveys for this study. We extend our gratitude to Anna Sparrman and Victoria Wibeck at the Department of Thematic Studies, Linköping University, for reading and commenting on a previous version of the paper. Thanks also to our previous student Sara Spjuth for conducting the interviews in one of the municipalities as part of her bachelor thesis work.

**Conflicts of Interest:** The authors declare no conflict of interest.

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
