# Peer review of "Integrating Sustainable Stormwater Management in Urban Planning: Ways Forward towards Institutional Change and Collaborative Action"

_water, doi:10.3390/w12010203_

Round 1

Reviewer 1 Report

Thank you for your manuscript. It is an interesting study about the current problems in Swedish stormwater management. Yet, it feels like the manuscript is a 'scientific translation' of a several interview results put together, mixed with some discussions of how institutional barriers could be improved.

The manuscript could be greatly improved by making clear which methods (interviews, workshops, questionnaires) were used for what purpose and which results were obtained by each method. 

In the 'results' section, please make a clear distinction between results, discussion and conclusions. It is hard for the reader to distinguish these.

What is your contribution to the scientific community? The conclusions and suggestions seem quite obvious and general. What is learned and new in the study?

The level of English is good.

More comments are put in the attached document.

Author Response

Dear Reviewer 1, 

Many Thanks for thoughtful and critical comments  tha have helped us in improoving this revised version of the manuscript. Please see an overview of our responses to your general comments attached with this message. As you also inserted commetns in the textfile you will find some of the answers in our revised version of the manuscript as commetns appear in the text. We are happy to make furter clarifications if needed. 

Kind regards 

The Authors

Reviewer 2 Report

The manuscript deals with the integration of sustainable stormwater management into urban planning. Empirical materials for the study come from a Swedish municipal context but the discussion on the study outcomes is addressed to a broader context.

The article is an original contribution and the topic is of interest for the readership of the Water journal.

English language is clear and the presentation is good; anyway, I have detected some criticisms in the text that should be properly addressed.

The Authors can benefit from the comments below to improve their paper. These have to be accomplished before manuscript acceptance.

Abstract

The abstract is concise and reflects the content of the article.

Introduction

Aims of the study are properly clarified in the Introduction.

Lines 28-31: concerning the intensification of problems related to conventional stormwater systems due to climate variation, the Authors are recommended to consider the following study as part of the introductory discussion:  

Todeschini S. (2012). Trends in long daily rainfall series of Lombardia (Northern Italy) affecting urban stormwater control. International Journal of Climatology, 32(6): 900-919, doi:1002/joc.2313.

Line 57: Replace “1” with “1”.

Background and theoretical departure

The section is clear and the provided references are adequate.

Materials and Methods

This section is clear but in my opinion not adequately detailed. I suggest the Authors to clarify how the data collected by means of workshops, questionnaires and interviews have been arranged/manipulated. How to treat a greater number of water utility officials than consultants (or municipal planners) in the total respondents to the proposed questionnaire?

Line 228: Replace “2” with “2”.

Line 249: remove the number of pages 171-172 after “24”.

Line 278-281: I suggest reporting a brief summary on the information acquired during the conferences and consultancies outside the research context.

Results and Analysis

Results are presented in a logical sequence. The proposed subdivision into subsections is effective. I suggest the Authors adding a schematic (perhaps a flow chart) of the detected criticalities/challenges and the proposed ways forward for the stormwater sector.

Sustainable stormwater management is an important issue that integrate and improve stormwatater control not an alternative to traditional stomwater control. Therfore, I suggest the Authors revising the sentence at lines 545-548 “there is less need for experts calculating dimensioning of pipes, but …” with “there is not only need for experts dimensioning of pipes, but …”.

Line 574: Remove “and others”.

Line 583: Section 4.1.2 or 4.1.3. Please, check.

Concluding discussion

The concluding discussion seems reasonable and is supported by the analysis presented in the previous sections.

Line 636: Replace “paperer” with “paper”. Typo error.

Line 648: Remove “For example”.

References

A reference is suggested in the “Introduction” Section concerning the enhanced problems on conventional urban drainage systems related to climate variation. Apart from this reference, based on my knowledge, no important reference is missing.

Author Response

Dear Reviewer 2, 

Many thanks for your comments which have helped us to improve this revised version of the manuscript. Please see an overview of our responses to your comments in the attached document. 

Kind regards 

The Authors

Round 2

Reviewer 2 Report

The manuscript has been significantly improved following the reccomandations of the Reviewers. All my concerns have been addressed and convingcly justified. In my opinion the paper can be accepted in the present form.